# Dietary Whey Protein Supplementation Increases Immunoglobulin G Production by Affecting Helper T Cell Populations after Antigen Exposure

**DOI:** 10.3390/foods10010194

**Published:** 2021-01-19

**Authors:** Dong Jin Ha, Jonggun Kim, Saehun Kim, Gwang-Woong Go, Kwang-Youn Whang

**Affiliations:** 1Division of Biotechnology, Korea University, Seoul 02841, Korea; djha7782@korea.ac.kr (D.J.H.); ketone@korea.ac.kr (J.K.); 2Division of Food Bioscience and Technology, Korea University, Seoul 02841, Korea; saehkim@korea.ac.kr; 3Department of Food and Nutrition, Hanyang University, Seoul 04763, Korea

**Keywords:** whey protein concentrate, immunity, rat

## Abstract

Whey protein is a by-product of cheese and casein manufacturing processes. It contains highly bioactive molecules, such as epidermal growth factor, colony-stimulating factor, transforming growth factor-α and -β, insulin-like growth factor, and fibroblast growth factor. Effects of whey protein on immune responses after antigen (hemagglutinin peptide) injection were evaluated in rats. Experimental diets were formulated based on NIH-31M and supplemented with 1% amino acids mixture (CON) or 1% whey protein concentrate (WPC) to generate isocaloric and isonitrogenous diets. Rats were fed the experimental diets for two weeks and then exposed to antigen two times (Days 0 and 14). Blood was collected on Days 0, 7, 14, and 21 for hematological analysis. The WPC group showed decreased IgA and cytotoxic T cells before the antigen injection (Day 0) but increased IgG, IL-2, and IL-4 after antigen injection due to increased B cells and T cells. Helper T cells were increased at Days 14 and 21, but cytotoxic T cells were not affected by WPC. WPC may activate adaptive immunity (IgG) against antigen by modulating helper T cells. Bioactive molecules might contribute to the immune-enhancing effects of whey protein concentrate.

## 1. Introduction

Living organisms are in constant competition for shared resources necessary for their survival. At the same time, they must protect themselves against threats or harm. All organisms have an innate immune system of defense, but only vertebrates possess T cells and the ability to produce antibodies. Innate immunity is defined as the first line of defense and is an immediate, non-specific response. Adaptive immunity is a highly specific response with immunological memory [1]. Adaptive immunity can be acquired by either natural (infection) or artificial (vaccination) exposure to an antigen. After primary antigen expose, the antigen-presenting cells interact with and influence the activation or suppression and differentiation of immature T cells into cytotoxic T cells (Tc cells) or helper T cells (Th cells). Tc cells kill infected or damaged cells, and Th cells regulate both innate and adaptive immune responses to a specific antigen via cell-mediated immunity and humoral immunity. Th cells activate and induce B cells to undergo clonal expansion into antibody-secreting plasma cells (humoral immunity). Some B cells and T cells become memory cells that rapidly differentiate into effector cells upon further antigen exposure and are responsible for long-term immunity by producing antibodies [2]. Antibodies (immunoglobulins, Igs) are classified into IgA, IgD, IgE, IgG, or IgM based on their biological properties, functional locations, and ability to manage different antigens [3]. IgG provides the majority of antibody-based immunity against pathogens and represents about 75% of the circulating antibody in humans [1]. Thus, non-pharmaceutical means to improve the host’s immune response after antigen exposure is an interesting area of study for protecting against harmful disease [4].

All living cells, including the immune system cells, require adequate and appropriate nutrients for optimal function. Inadequate nutrition causes defects in immune system development in infants and declined immune function in the elderly [5,6]. Dietary supplementation with trace minerals, antioxidants, vitamins, essential fatty acids, and probiotics is reported to modulate cellular and humoral immune responses [7,8,9]. Protein is the major structural component of all cells in the body [10]. Dietary protein provides essential amino acids for protein synthesis, as well as is important for satiety, energy metabolism, blood pressure, bone metabolism, and immune function [11], but dietary protein requirement in healthy individuals is traditionally defined as the lowest protein intake sufficient to achieve neutral body protein balance [12]. Allowance of protein for a healthy adult with minimal physical activity is currently 0.8 g protein per kg of body weight (BW) and dietary intake of 1.0, 1.3, and 1.6 g protein per kg BW per day is recommended for individuals with minimal, moderate, and intense physical activity, respectively [13]. Maternal protein energy malnutrition is one of the main causes of intrauterine growth retardation and also causes impairment of both cell-mediated and humoral immunity [14,15]. Pups born from rats fed a protein-deficient diet during pregnancy showed deficient humoral immune responses, diminished antibody titers, and decreases in the number of antibody-forming cells (plasma B-cells) after antigen exposure [16]. Older women with low protein intake have reduced lean mass and muscle function and reduced immune responses [17]. Dietary protein and certain amino acids are important for adequate immune function and immunomodulation in the organism [18,19]. In addition, dietary protein supplementation is known to boost immune competence [20,21].

Among the dietary protein sources, bovine milk contains easily absorbable minerals and proteins compared to cereal proteins. Whey protein is a by-product of cheese and casein manufacturing processes but contains highly bioactive molecules, such as epidermal growth factor (EGF), colony-stimulating factor (CSF), transforming growth factor (TGF)-α and -β, insulin-like growth factor (IGF), and fibroblast growth factor (FGF) with high quality protein [22,23]. Some bioactive molecules, such as lactoferrin, EGF, and TGF-β, resist enzyme digestion, while others can be produced by enzymatic hydrolysis of whey protein in vivo and in vitro [11,24,25]. Bioactive peptides exert beneficial health effects by affecting metabolism (glucose, fat, and amino acids) [26,27,28,29]. Moreover, whey protein extract stimulated neutrophils and lymphocytes [30,31,32], and feed containing fermented whey protein lowered the levels of infiltrating neutrophils in atopic contact dermatitis in mice [33]. The most abundant protein in whey protein is β-lactoglobulin, which stimulates the proliferation of spleen cells and lamina propria lymphocytes [34,35,36]. The second most abundant protein in whey protein, α-lactalbumin, modulates macrophages and B and T cell functions in ruminants [37]. Although whey protein has beneficial effects on human health [38], abusive use of whey protein as protein supplement (about 40 g per day) has potentially adverse effects on liver, kidney, and intestinal microbiota [39]. However, relatively little research has been conducted on the efficacy of whole protein supplements as a dietary immune-modulator.

In this study, we tested the immunomodulatory effect of whey protein concentrate (WPC) in rats exposed and re-exposed to hemagglutinin peptide as an antigen to determine the adaptive immune response. The humoral immune response was determined by measuring the plasma Ig concentration and cytokine levels. The cell-mediated adaptive immune response was evaluated by measuring the B cell and T cell populations. The immune-enhancing effects of WPC diet, including bioactive molecules, were compared to amino acids-supplemented diet.

## 2. Materials and Methods

### 2.1. Dietary Treatment

WPC (Alacen 878) containing 79.5% crude protein, 10% crude fat, 8.0% lactose, 0.78% Ca, and 0.2% P was obtained from New Zealand Milk Products, Inc. Based on the NIH-31M diet, the control and WPC diet were formulated to contain 10 g of the amino acids mixture per 1 kg of feed (group CON) and 10 g of WPC per 1 kg of feed (group WPC) (Table 1). The amino acids mixture was formulated based on the nutrients composition of WPC to generate isocaloric and isonitrogenous diets with supplementation of essential amino acids.

### 2.2. Animal Experiment

As estrus cycle of female has shown hormonal effects on immunity [40,41], we randomly assigned 6-week-old male Sprague-Dawley (SamTacN(SD)BR) rats (Samtako, Gyung-gi, Korea) into two dietary treatments (20 rats per treatment). During the experiment, rats were housed in individual stainless-steel, wire-bottom cages and kept in a temperature- and light-controlled room (22 ± 2 °C and 12-h day-night cycle) in the Small Animal Experimental Unit (Korea University, Seoul, Korea). Rats had free access to water and feed throughout the experiment. After 1 week of acclimatization, the rats were fed either the control diet (CON) or the whey protein concentrate containing diet (WPC) for 14 days before antigen injection. Influenza hemagglutinin peptide (15 μg/kg body weight [BW]; Sigma-Aldrich, St Louis, MO, USA) as the antigen was administered by intramuscular injection, and a second antigen exposure was conducted 14 days after the first injection. The day of the first antigen injection was designated as Day 0. BW and feed intake (FI) were measured weekly, and daily weight gain and daily FI were calculated. On Days 0, 7, 14, and 21, whole blood (5 rats/treatment) was collected by cardiocentesis under anesthesia (intraperitoneal injection of ketamine at 90 mg/kg BW and xylazin at 10 mg/kg BW) using an EDTA-containing vacutainer (BD Bioscience, San Diego, CA, USA) and then rats were euthanized.

### 2.3. Hematological Analysis

Anticoagulant (EDTA)-treated whole blood was collected to determine red blood cells, platelets, white blood cells, lymphocytes, neutrophils, basophils, and eosinophils (Green Cross Reference Lab, Kyungki, Korea).

Plasma was obtained by centrifugation, and IgG (Koma-Biotech, Seoul, Korea), IgA (Koma-Biotech), interleukin (IL)-2 (R&D systems, Minneapolis, MN, USA), IL-4 (BD Biosciences), tumor necrosis factor-α (TNF-α, BD Biosciences), and interferon-γ (INF-γ, R&D Systems) were measured according to the manufacturers’ protocols.

Red blood cells were lysed with Pharm-Lyse^TM^ (BD Biosciences), and lymphocytes were collected by centrifugation (200× *g* at 4 °C for 5 min). Collected lymphocytes were resuspended in PBS containing 1% fetal bovine serum and incubated with antibodies (against CD3, CD4, CD8, and CD45r) conjugated with florescent (BD Biosciences). Lymphocyte populations were measured with a FACSCalibur flow cytometer (BD Biosciences).

### 2.4. Statistical Analysis

All results were analyzed using SAS (SAS Institute, Inc., Cary, NC, USA). Differences between dietary treatments were determined by the Student *t*-test. Changes over time within dietary treatments were assessed by one-way ANOVA, followed by multiple comparisons based on Fisher’s least significant difference test. Differences were considered significant at *p* < 0.05, and 0.05 < *p* < 0.10 for numerical differences.

## 3. Results

### 3.1. Dietary Treatments Did Not Affect the Body Weight (BW) and Feed Intake (FI)

BW and FI were not affected by dietary treatments (Figure 1). After antigen injection (Day 0 and 14), the rats showed a decrease in BW gain compared to the previous week due to the decrease in FI in both dietary treatments (*p* = 0.04).

### 3.2. Dietary WPC Changed the White Blood Cells (Neutrophil and Lymphocyte) Population after Antigen Injection

The blood cell populations are presented in Table 2. The populations of red blood cells and platelets were not affected by dietary treatments or antigen injection throughout the experiment. Before antigen injection, dietary treatments did not affect the white blood cell counts, but, after antigen exposure, the white blood cells increased numerically in both dietary treatments (*p =* 0.07). The percentage of neutrophil in white blood cells was not changed by dietary treatments before antigen injection but decreased after antigen injection, and these decreases differed between the dietary treatments. Group WPC showed significant decreases in the neutrophils after Day 7, and further decreases were found on Days 14 and 21. Group CON showed no difference on Day 7, but significant decreases afterward (Days 14 and 21; *p* < 0.04). Compared to group CON, group WPC showed greater decreases in neutrophils, and these decreases were significant (*p =* 0.03). Other granulocytes accounted for less than 3% of white blood cells and did not show any changes due to dietary treatment or antigen injection, or both. Dietary treatments alone did not affect the percentage of lymphocytes (Day 0), but both groups (WPC and CON) changed differentially after antigen injection. Group WPC showed numerically increased lymphocytes on Day 7 and significant increases on Days 14 and 21, while group CON showed numerical increases on Days 14 and 21. No significant differences in lymphocyte proportion were found until Day 14, and the lymphocyte proportion was significantly less in group WPC than in group CON on Day 21 (*p =* 0.05).

### 3.3. Dietary WPC Increased B Cell and Helper T Cell Populations after Antigen Injection

We determined the lymphocyte subpopulation (B cells and T cells) by using the cluster of differentiation (CD) markers (Table 3 and Appendix A). B cells (CD45R+ cells) were not affected by dietary treatment without antigen injection (Day 0) but increased after antigen injection in both the CON and WPC groups. The B cells was increased numerically in group CON, while WPC treatment showed significantly increased B cells 14 days after antigen injection (*p* = 0.04, Day 14), and this increase was significantly higher compare with group CON on Day 21 (seven days after the second antigen injection; *p =* 0.05). T cells (CD3+ cells) also showed no difference by dietary treatment alone (Day 0). Group CON showed no difference in T cells even after antigen injection. By contrast, group WPC showed a significant increase from Day 7 and significantly more T cells than group CON on Day 21 (*p* = 0.04).

The subpopulations of T cells (CD3+), Tc cells (Appendix A), and Th cells (Appendix A) were distinguished by using CD8 and CD4 markers, respectively. WPC treatment before antigen injection (Day 0) showed a decrease in Tc cells compared to CON treatment (*p =* 0.04). After antigen injection, Tc cells in both dietary treatments showed no change even after the second antigen injection on Day 14. Dietary treatments did not affect the Th cells (Day 0), but, after antigen injection, Th cells increased gradually in group CON (*p =* 0.07) and increased significantly in group WPC (*p* = 0.05). WPC treatment increased the Th cells significantly compared to group CON at seven days after antigen injection (Days 7 and 21). This increment contributed to the increased T cell populations in group WPC on Day 21 (*p* = 0.05).

### 3.4. Dietary WPC Increased Blood IgG and Cytokines (IL-2 and IL-4) after Antigen Injection

The adaptive immune response involves B cell-mediated humoral and T cell-mediated cellular immune responses [1]. B cells produce Igs, and this process requires specific molecules produced from other immune cells. WPC changed the lymphocyte populations, especially B cells and Th cells, after antigen injection (Table 3). We also determined humoral immunity by measuring the plasma IgG and IgA levels and cytokines (IL-2, IL-4, TNF-α, and INF-γ). Plasma IgG was not affected by dietary treatment alone (Day 0, Figure 2) but elevated after antigen injection (Days 7, 14, and 21) in both the CON and WPC groups (*p* = 0.04). Comparing the dietary treatments, only a significantly higher IgG content was found in WPC treatment on Day 21 (*p* = 0.03). Interestingly, WPC dietary treatment lowered IgA before antigen injection (Day 0) (*p* = 0.05), and this diminished level of IgA was elevated by antigen injection (Days 7, 14, and 21). Group CON showed significantly increased IgA only on Day 14, and no difference was found between dietary treatments (Days 7, 14, and 21). Pretreatment with WPC did not affect IL-2, IL-4, TNF-α, and INF-γ. Antigen injection stimulated the production of IL-2, IL-4, TNF-α, and INF-γ in both dietary treatments (*p* = 0.04), but the IL-2 level in group CON and TNF-α level in group WPC dropped to the initial level on Day 21. WPC increased IL-2 and IL-4 on Day 14 compared to CON, and there was no difference between the effects of the dietary treatments at other time points.

## 4. Discussion

We determined the effects of WPC on the immune responses in rats after antigen injection. Hematology parameters, including lymphocyte populations and humoral immunity, were determined. Rats showed a decreased FI after antigen injection (hemagglutinin). This stress-inducing procedure might have decreased the appetite of rats. Vaccination (antigen exposure) triggers the immune system, which activates innate and acquired immunity and can lead to adverse side effects, including loss of appetite [42]. At the injection site, the rapid response of innate immunity causes immediate inflammation by producing pro-inflammatory cytokines and recruitment of several types of cells (phagocytic and non-phagotytic) to prevent foreign substances from spreading [1]. The reduced FI caused by antigen injection indicates successful activation of immune system response.

The immune system can be divided into two components: the innate immune system and the adaptive immune system. The innate immune system is a non-specific response to foreign substances and is conducted by the complement system and leukocytes (neutrophils, eosinophils, basophils, and monocytes). Due to their low cell numbers, we did not observe any changes in these leukocytes, except for a decreased in neutrophils in rats fed WPC before antigen injection. Neutrophils are phagocytic cells that are involved in bacterial infection [43]. The decreased neutrophil population following dietary treatment with WPC might be due to decreased gastrointestinal track exposure (GIT) to environmental bacteria. Commensal bacteria in the GIT are prevented from causing infection by the barrier function [44,45]. This gut barrier function is regulated by tight junction protein, claudins, that modulate the permeability of tight junctions [46]. WPC increased claudin expression in HT-29 cells by activating the TGF-β receptor [47]. This improved gut barrier function was confirmed in piglet model challenged with lipopolysaccharide [48]. We did not measure the serum lipopolysaccharide concentration as a marker of bacterial infection. However, increased gut barrier function via WPC supplementation might decrease pathogenic infection in the GIT, in turn, decreasing innate immunity (neutrophils proportion).

Similar to the trend of decreased neutrophils by WPC treatment before antigen injection, plasma IgA and Tc cell populations were also decreased (*p* < 0.05). IgA is the most abundant Ig in the human body. It is secreted in treat, saliva, sweat, colostrum, and mucus and responsible for mucosal immunity. Only a small amount IgA is found in blood as a soluble form of IgA [49]. Most IgA is found in the mucus membrane, where it plays a crucial role in the mucous membrane’s immune function [3]. Increased bacterial exposure in the gut is known to stimulate IgA production [50]. In addition, intestinal epithelial cells express major histocompatibility complex (MHC) class II molecules, which induces the development of Th cells to Tc cells in response to pathogen exposure [51]. Decreased Tc cells with increased Th cells in WPC treatment might indicate decreased MHC class II presentation due to increased gut barrier function.

Adaptive immunity, particularly involving B cells and T cells, was examined by measuring the lymphocytes subpopulation (Table 3). Fourteen days of pre-dietary treatment did not affect the B cell population (on Day 0) but numerically decreased the Tc cells. Group CON showed a gradual increase in the B cell population after antigen injection, while group WPC showed a significant increase after Day 14. B cells are a component of the adaptive immune system and are especially involved in humoral immunity by secreting antibodies (Igs), making antigens, and secreting cytokines [52]. Th cells stimulate the maturation of B cells into plasma cells that produce Igs. Plasma B cells express the surface protein CD40L and secrete cytokines (such as IL-4 and IL-21), which promote B cells proliferation, Ig class switching, somatic hypermutation, and sustain T cell growth and differentiation [1,53]. The increase in B cells by WPC after 14 days of antigen exposure led to increased plasma IgG on Day 21, which indicates that WPC is effective for stimulating antibody production after re-exposure to the antigen.

IL-2 is a cytokine produced by antigen-activated T cells. It promotes clonal expansion of antigen-specific T cells [54,55], and the differentiation of T cells into effector T cells (Th cells) and memory T cells [56]. The elevated plasma level of IL-2 in group WPC on Day 14 might sustain the increased population of Th cells. IL-4 showed a pattern similar to IL-2. IL-4 is a B cells activator and differentiation factor that regulates Ig isotype switching, particularly that of IgG_1_ [57]. We did not measure plasma B cells, which are non-proliferating antibody-secreting cells arising from B cell differentiation [1]. However, increased plasma IgG on Day 21 might be due to the increased population of plasma B cells by IL-2 and IL-4 stimulations.

TNF-α was increased by antigen injection, but no difference was found between the dietary treatments (CON vs. WPC). Group CON showed an elevated level of TNF-α on Day 21, but group WPC showed a diminished level of TNF-α, and this level was similar to that on Day 0 (before antigen injection). TNF-α is produced by activated macrophages, which engulf and digest cellular debris, foreign substances, and microbes by phagocytosis. A diminished level of TNF-α on Day 21 indicates rapid clearance of injected antigen by activated macrophages. We expected an elevated INF-γ level in group WPC, but no difference was found after antigen injection compared to group CON. INF-γ is a critical cytokine that affects innate and adaptive immunity by immunostimulatory and immunomodulatory effects [1] besides inducing various immune response against viral or microbial pathogens [58]. In this experiment, we employed HA peptide, which did not have any viral activity; this might not stimulate INF-γ production from the activated macrophage.

Dietary whey protein concentrate supplementation increased the B cell population via Th cell-mediated increases in IL-2 and IL-4 levels and increased humoral immunity, particularly plasma IgG production. Moreover, WPC decreased the IgA and Tc cell levels before antigen challenge. We hypothesize that dietary whey protein concentrate may prevent antigen exposure by improving innate immunity and also activate adaptive immunity when antigen exposure is increased. Its effect seems not to be just increased nutrient (amino acids) supplementation because both diets were isocaloric and isonitrogenous. In addition to bioactive molecules, such as EGF, CSF, TGF-α, TGF-β, IGF, and FGF, which show stability against digestive enzymes and bioactive peptides produced by enzymatic digestion of whey protein [11,24,25] play important roles in the immune-enhancing effects of whey protein concentrate.

## 5. Conclusions

We determined the immune-enhancing effects of whey protein concentrate after antigen exposure in rats. Proteins are an essential nutrient and protein requirement is defined as the lowest intake level sufficient to achieve neutral body protein balance. Among the protein sources for human, animal foods show higher protein quality compared to plant source for providing the proteins in correct ratio of amino acids. After enzymatic digestion in digestive track, protein could produce bioactive peptides and bioactive molecules, such as EGF, CSF, TGF-α and -β, IGF, and FGF, might have additional beneficial effects on health. As a result of this study, WPC improved the immune response compared to amino acids supplementation, which indicates that the protein can offer more health benefits than amino acids. Increased whey protein consumption might increase the immune response against antigen exposure and might decrease the incidence of disease. The molecular mechanisms of the immune-enhancing effect of whey protein bioactive molecules should be addressed in the future.

## Figures and Tables

**Figure 1 foods-10-00194-f001:**
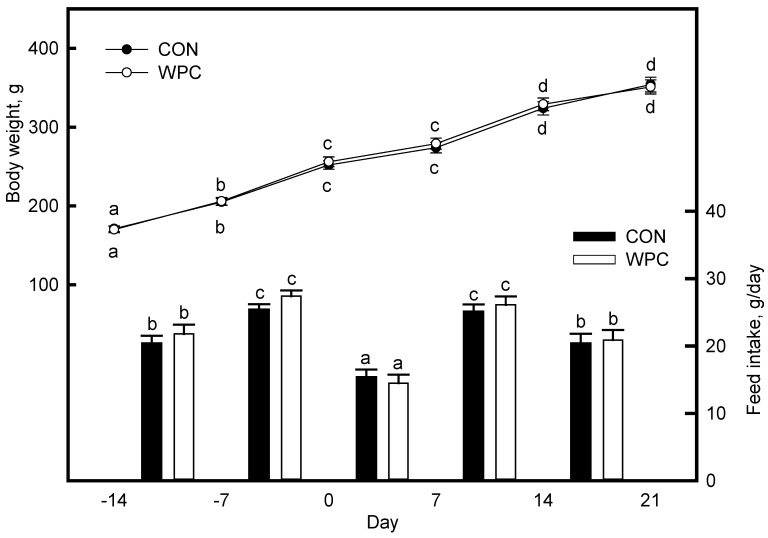
Body weight and feed intake of rats fed the experimental diets for 35 days. Day 0 is the first antigen exposure day by injection. Values are expressed as mean ± S.E. (*n* = 5). ^a,b,c^^,d^ Significantly different compared with Day 14 of each dietary treatment by ANOVA and Fisher’s least significant difference test (*p* < 0.05).

**Figure 2 foods-10-00194-f002:**
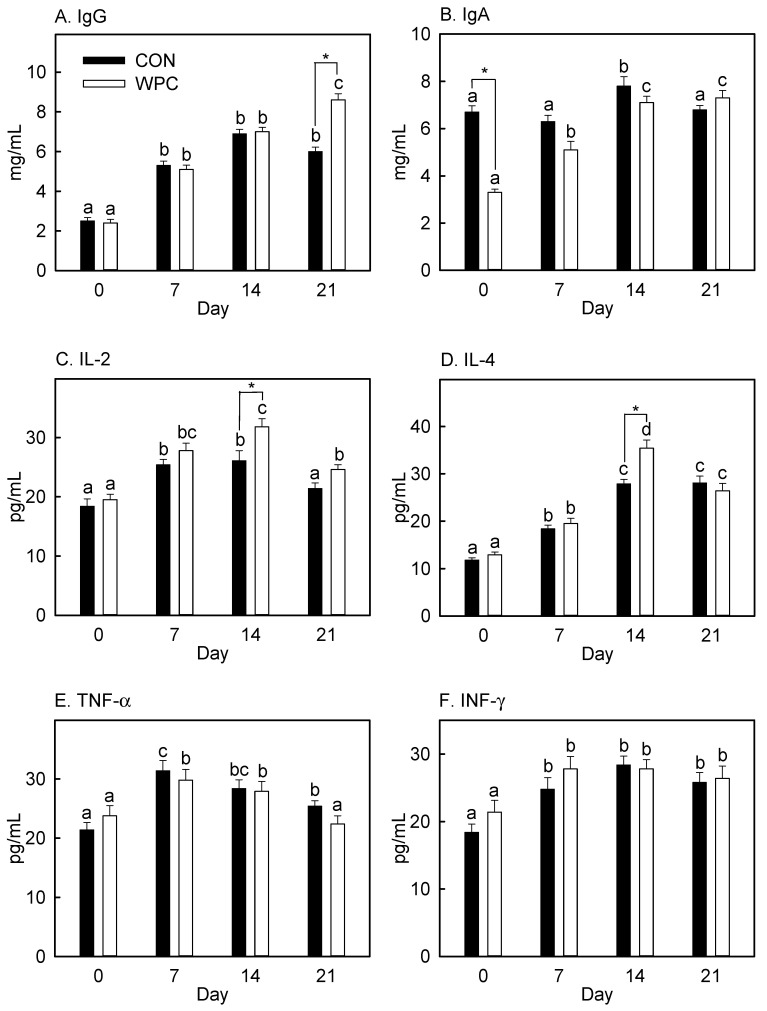
Changes in plasma immunoglobulins and cytokines concentrations in rats after antigen injection. Values are expressed as mean ± S.E. (*n* = 5). IgG, immunoglobulin G; IgA, immunoglobulin A; IL-2, interleukin-2; IL-4, interleukin-4; TNF-α, tumor necrosis factor-α; INF-γ, interferon-γ. * Significantly different when compared with CON on an identical Day by Student *t*-test (*p* < 0.05). ^a,b,c^^,d^ Significantly different compared with Day 0 of each dietary treatment by ANOVA and Fisher’s least significant difference test (*p* < 0.05).

**Table 1 foods-10-00194-t001:** Composition of CON and WPC diets in the experiment (g/kg).

Ingredient	CON	WPC
Amino acids mixture ^1^	10.0	-
Whey protein ^2^	-	10.0
Ground whole wheat	341.7	341.7
Ground whole yellow corn	200.0	200.0
Ground whole oats	120.0	120.0
Wheat middlings	100.0	100.0
Fish meal	90.0	90.0
Soybean meal	50.0	50.0
Soybean oil	25.0	25.0
Alfalfa meal	20.0	20.0
Corn gluten meal	20.0	20.0
Dicalcium phosphate	15.0	15.0
Brewer’s dried yeast	10.0	10.0
Ground limestone	5.0	5.0
Salt	5.0	5.0
Vitamin premixture ^3^	2.5	2.5
Mineral premixture ^4^	2.5	2.5
Choline chloride	1.3	1.3
L-Lysine	1.0	1.0
DL-Methionine	1.0	1.0

^1^ Amino acids mixture (g/100 g): His 1.7, Ile 4.5, Leu, 8.4, Lys 7.5, Met+Cys 3.3, Phe 2.6, Thr 5.6, Trp 1.6, Val 4.5, Glu 39.6, lactose 8, Ca 0.78, P 0.2, corn oil 10, and cellulose 3.53. ^2^ Whey protein concentrate (Alacen 878) from New Zealand Milk Products, Inc. ^3^ Vitamin Premixture (per kg diet): 24,300 IU vitamin A, 4200 IU vitamin D_3_, 22 mg vitamin K, 16.5 mg vitamin E, 132 µg biotin, 1.1 mg folic acid, 22 mg niacin, 27.5 mg pantothenic acid, 2.2 mg pyridoxine, 5.5 mg riboflavin, 71.5 mg thiamine, 15.4 μg vitamin B12. ^4^ Mineral premixture (per kg diet); 440 μg Co 4.4 mg Cu 66 mg Fe, 440 mg Mg, 110 mg Mn, 11 mg Zn, 7.7 mg I.

**Table 2 foods-10-00194-t002:** Hematology parameters of rats fed experimental diets after antigen injection.

Variable	Day 0		Day 7		Day 14		Day 21	
	CON	WPC	CON	WPC	CON	WPC	CON	WPC
Red blood cells, 10^6^ cells/µL	8.8 ± 0.6	8.7 ± 0.5	8.9 ± 0.4	8.8 ± 0.4	8.2 ± 0.5	8.5 ± 0.5	8.3 ± 0.4	8.3 ± 0.5
Platelets, 10^6^ cells/µL	1.1 ± 0.2	1.1 ± 0.3	1.3 ± 0.2	1.2 ± 0.2	1.2 ± 0.3	1.1 ± 0.2	1.2 ± 0.1	1.4 ± 0.2
White blood cells, 10^3^ cells/µL	8.9 ± 0.8	7.3 ± 0.9	11.8 ± 1.1	12.4 ± 1.0	9.8 ± 0.7	12.4 ± 0.8	13.1 ± 1.1	12.8 ± 1.0
Neutrophils, %	21.0 ± 1.6 ^a^	21.0 ± 2.2 ^a^	21.7 ± 1.8 ^a^	17.9 ± 1.1 *^,b^	17.4 ± 1.1 ^b^	12.0 ± 0.8 *^,c^	17.8 ± 1.4 ^b^	10.1 ± 0.7 *^,c^
Lymphocytes, %	76.9 ± 2.4	76.6 ± 2.8 ^a^	76.5 ± 2.1	80.0 ± 2.2 ^a^	80.2 ± 2.1	85.2 ± 2.1 ^b^	79.3 ± 1.8	88.2 ± 2.2 *^,b^
etc. ^1^, %	2.1 ± 0.4	2.4 ± 0.5	1.8 ± 0.2	2.1 ± 0.4	2.4 ± 0.5	2.8 ± 0.4	2.9 ± 0.4	1.8 ± 0.3

Values are expressed as mean ± S.D. (*n* = 5). * Significantly different when compared with CON on an identical Day by the Student *t*-test (*p* < 0.05). ^a,b,c^ Significantly different when compared with Day 0 of each dietary treatment by ANOVA and Fisher’s least significant difference test (*p* < 0.05). ^1^ etc., the sum of eosinophils, basophils, and monocytes.

**Table 3 foods-10-00194-t003:** Changes in B cell and T cell populations in rats after antigen injection.

	CON	WPC
B cells (CD45R+), %		
Day 0	20.6 ± 1.8	22.3 ± 1.2 ^a^
Day 7	21.4 ± 1.1	20.8 ± 1.8 ^a^
Day 14	23.0 ± 1.8	26.0 ± 1.9 ^b^
Day 21	24.0 ± 2.1	28.1 ± 3.1 *^,b^
T cells (CD3+), %		
Day 0	56.3 ± 1.2	54.3 ± 1.3 ^a^
Day 7	55.1 ± 1.8	59.2 ± 2.1 ^b^
Day 14	57.2 ± 1.8	59.2 ± 2.4 ^b^
Day 21	55.3 ± 2.1	60.1 ± 1.8 *^,b^
Cytotoxic T cells (CD3+/CD8+), %		
Day 0	25.3 ± 0.8	22.4 ± 0.6 *
Day 7	22.8 ± 1.8	22.8 ± 1.4
Day 14	20.4 ± 1.4	20.9 ± 1.7
Day 21	20.8 ± 1.5	20.7 ± 1.4
Helper T cells (CD3+/CD4+), %		
Day 0	31.0 ± 1.8	31.9 ± 1.2 ^a^
Day 7	32.3 ± 1.1	36.4 ± 1.9 *^,b^
Day 14	36.8 ± 2.1	38.3 ± 1.8 ^b^
Day 21	34.5 ± 1.3	39.4 ± 1.5 *^,b^

Values are expressed as mean ± S.D. (*n* = 5). * Significantly different when compared with CON on an identical Day by Student *t*-test (*p* < 0.05). ^a,b^ Significantly different compared with Day 0 of each dietary treatment by ANOVA and Fisher’s least significant difference test (*p* < 0.05).

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
