# Peer review of "Dietary Whey Protein Supplementation Increases Immunoglobulin G Production by Affecting Helper T Cell Populations after Antigen Exposure"

_foods, 2021, doi:10.3390/foods10010194_

Round 1

Reviewer 1 Report

The study entitled ‘Dietary whey protein supplementation increases immunoglobulin G production by affecting helper T cell populations after antigen exposure’ by Ha et al., looks interesting but experiments are not executed well and poorly written. The following issues should be addressed for further consideration of the manuscript.

  1. The title itself not clear whether dietary whey protein supplementation or antigen exposure increases immunoglobulin G production or affecting helper T cell populations? The aim of the study also clearly explained
  2. The introduction seems very short. Give more information about immunological functions of immunoglobulin G production and helper T cells. Why this study is important in the current scenario? Provide adverse effect of dietary whey protein on human health.3
  3. Reference should be cited from recent publications. Most of the references cited in this manuscript are old. I recommend to include recent publications in this manuscript. That will fine if cite both old and recent references. The recent publication will give the current status of research related to Dietary whey protein supplementation and immunological responses
  4. The result should be a concise and more clear explanation. Add subheading in results with 3.1., 3.2., etc. for clear understanding. Without subheading difficult to understand for readers. Every subheading explains the findings of the results.
  5. Table 2. Insisted on the table, represent the body weight in the graph.
  6. The author mentioned that lymphocyte subpopulation (B cells and T cells) was measured by FACSCalibur (BD bioscience). Provide the pictures obtained from FACSCalibur for CD45R positive cells and CD3 positive cells for all groups.
  7. Changes in plasma immunoglobulins and cytokines represent in a bar diagram with statistical value.
  8. The manuscript required language editing by a native English speaker.

Reviewer 2 Report

Manuscript ID: foods-1028898
Type of manuscript: Article
Title: Dietary whey protein supplementation increases immunoglobulin G
production by affecting helper T cell populations after antigen exposure
Authors: Dong Jin Ha, Jonggun Kim, Saehun Kim, Gwang-woong Go *, Kwnag-Youn
Whang *

Summary

The author is conducting a hematological analysis in rats with whey protein concentrate supplemented diet after antigen exposure. The research is well designed. With only 1% WPI supplementation and only 5 animals per timepoint the described hematological analysis effects are impressive. However, there are some major and minor concerns.

Comments

Major

Statistics: Are poor. Only done with student-t test. No ANOVA is used to study time effect and correction of multiple comparisons are missing.

There is a substantial amount of text in abstract and introduction about potential bio-active molecules in the whey protein based on vitro studies, but there is no background information if these molecules are still available after digestion in vivo and in what form/amount they can be detected. Also, in this manuscript these molecules are not measured in systemic blood. It was also not further discussed in extension in the discussion section. Which is correct but it comes back in the last few sentences of the discussion section and abstract. I suggest to revise.

Innate immunity: There is extensive discussion about improved gut barrier function, but is based only on the neutrophil measurement and a lot of literature. Gut barrier function is not measured. Therefore, the conclusion in the abstract need to be revised

Why was this research only done in male rats?

How translational is this to humans?

Minor

Abstract: Line 17 describes two experimental groups “CON“ and “WP”. Unclear what group “WPC”  stands for.

M+M

Material and Method: The supplements are well described, but the contents of the rest of the food is missing.

Statistics: P-values need to more precise. Not “p<0.01” but “p=…..”

Results:

Table 2: In the result section, It is mentioned that there are no differences. However, how is tested is unclear. Please add statistical information to the table. Add number (N) of observations to the table

Table 3+4+5: Add number (N) of observations to the table,

Discussion line 258: “conclude” should be “hypothesize”

Reviewer 3 Report

The assumption is interesting, but its description is questionable. The introduction is not based on the latest literature on this topic. The authors introduced whey protein in the form of a protein concentrate to the diet of rats. However, they did not provide the most important thing - how many whey proteins were there actually? Whey protein preparations (WPC) contain 35 to 80% of whey proteins on a dry matter basis. The rest of the dry matter is mainly lactose. This can make a big difference to your conclusions. This must be completed in this article.

Detailed comments

L1. I propose to change fascinating to interesting

L41 Give protein whey instead of whey

L30-53 Has this topic not been raised by scientists for the last 10 years? Articles from 1974-2010 were used here.

L58 Give the exact composition of the WPC, especially its protein content. Whey protein concentrates contain 35 to 80% protein and even more. The experiment cannot be correctly interpreted if the actual amount of whey protein and how many other components of the whey, e.g. lactose, is not stated. How was the dry mass of the concentrate? Without these messages, the experience becomes meaningless and conclusions cannot be seen.

L86 Write down why the feeding lasted only 14 days, is it enough? Did you suggest yourself?

Table 2 If bodyweight and feed intake were not affected by dietary treatments - then enter SEM (standard error of the mean) in the table

L187-189 This passage is a repetition, it has already been written

L197 To be explained

L210-213 Based on what do you think? It needs more explanation. It is a very complex mechanism.

L223-227 This should be in the results.

L255-266 This summary does not add anything new and nothing constructive. It is a repetition of what was previously described in this manuscript.

L267 The conclusion is missing. The authors did not provide specific conclusions from this experiment

L277 Journal titles are incorrect. Authors did not follow the guidelines for authors.

Round 2

Reviewer 1 Report

The manuscript is much improved than the previous version and the authors carried out the corrections what I suggested. I appreciate the authors for their effort to make this manuscript. However, few corrections will be needed to increase the significance of the manuscript.

Minor corrections:

  1. Insisted of giving headings in the results section like this, pls try to give results of each section as a heading.

3.1. Body weight (BW) and feed intake (FI)

3.2. Hematology parameters

3.3. Lymphocyte subpopulation

3.4. Blood Igs and cytokines

For example, 3.1. dietary treatments did not affect the body weight (BW) and feed intake (FI)

  1. Supplementary Figure S1, S2 and S3: Check the spelling “Representive”. Add more description in the supplementary legends.

Author Response

Dear Reviewer,

Thank for your care review.

We changed sub-headings in the results from criteria to simple description of results

3.1. Body weight (BW) and feed intake (FI) ==>

Dietary treatments did not affect the body weight (BW) and feed intake (FI)

3.2. Hematology parameters ==>

Deitary WPC changed the white blood cells (neutrophil and lymphocyte) poplulation after antigen injection

3.3. Lymphocyte subpopulation ==>

Dietary WPC increased B cell and helper T cell populations after antigen injection

3.4. Blood Igs and cytokines ==>

Dietary WPC increased blood IgG and cytokines (IL-2 and IL-4) after antigen injection

and also, we added more description in supplemented Figure about the abbreviations (CON, WPC, Day, and CD markers).

The changes were colored with blue.